# Pseudo-Static Gain Cell of Embedded DRAM for Processing-in-Memory in Intelligent IoT Sensor Nodes

**DOI:** 10.3390/s22114284

**Published:** 2022-06-04

**Authors:** Subin Kim, Jun-Eun Park

**Affiliations:** Department of Electronics Engineering, Chungnam National University, Daejeon 34134, Korea; tnqls3409@o.cnu.ac.kr

**Keywords:** processing-in-memory (PIM), gain-cell embedded DRAM (eDRAM), pseudo-static leakage compensation (PSLC)

## Abstract

This paper presents a pseudo-static gain cell (PS-GC) with extended retention time for an embedded dynamic random-access memory (eDRAM) macro for analog processing-in-memory (PIM). The proposed eDRAM cell consists of a two-transistor (2T) gain cell with a pseudo-static leakage compensation that maintains stored data without charge loss issue. Hence, the PS-GC can offer unlimited retention time in the same manner as static RAM (SRAM). Due to the extended retention time, bulky capacitors in conventional eDRAM are no longer needed, thereby, improving the area efficiency of eDRAM-based analog PIMs. The active leakage compensation of the PS-GC can effectively hold stored data even in a deep-submicron process that show significant leakage current. Therefore, the PS-GC can accelerate write-access time and read-access time without concern of increased leakage current. The proposed gain cell and its 64 × 64 eDRAM macro were implemented in a 28 nm CMOS process. The bitcell of the proposed gain cell has 0.79- and 0.58-times the area of those of 6T SRAM and 8T STAM, respectively. The post-layout simulation results demonstrate that the eDRAM maintains the pseudo-static operation with unlimited retention time successfully under wide range variations of process, voltage and temperature. At the operating frequency of 667 MHz, the eDRAM macro achieved an operating voltage range from 0.9 to 1.2 V and operating temperature range from −25 to 85 °C regardless of the process variation. The post-layout simulated write-access time and read-access time were below 0.3 ns at an operating temperature of 85 °C. The PS-GC consumes a static power of 2.2 nW/bit at an operating temperature of 25 °C.

## 1. Introduction

In recent years, studies have extensively investigated energy-efficient hardware for deep neural network (DNN) applications. One such approach is the processing-in-memory (PIM) or computing-in-memory architecture that realizes efficient data processing to overcome the memory-processor bottleneck. A low-power and compact implementation of a DNN with PIM can be used to realize intelligent Internet-of-Things (IoT) sensor nodes, as illustrated in Figure 1. 

Many studies have implemented PIM based on static random-access memory (SRAM) due to its logic compatibility and high operation speed [1,2,3,4,5,6,7]. However, SRAM-based PIMs have the limitations of low bit density and large silicon area [1,2]. As an alternative, some studies have adopted next-generation nonvolatile memories, such as resistive RAM [8,9] or phase-change RAM [10]; however, these cannot easily be employed in general CMOS processes, and they require additional process steps that increase the manufacturing cost.

To overcome these limitations, PIMs with embedded dynamic random-access memory (eDRAM) have been proposed [11,12,13]. Logic-compatible eDRAMs [14,15,16,17] can offer a higher bit density and smaller area than those of the SRAMs. Hence, the eDRAM-based PIM can realize more area-efficient implementation than that of the SRAM-based PIM. In addition, eDRAM can be implemented in any CMOS process without the use of an additional layer. 

Unfortunately, eDRAM has a finite data retention time, and it requires a periodic refresh to maintain the stored data [18,19]. Previous eDRAM structures extended the retention time by employing an additional capacitor in the gain cell. However, the multiply-accumulate (MAC) operation in an analog PIM usually requires metal–oxide–metal (MOM) coupling capacitors [3,4,5], and a sufficiently large capacitor cannot be employed in the gain cell because of the area constraint. In addition, for the same gain cell architecture, process scaling to the ultra-deep submicron scale further reduces the retention time. 

As shown in Figure 2, for the same two-transistor (2T) gain cell structure [20,21], the simulated retention time decreases by approximately 300 times as the channel length decreases from 180 to 28 nm owing to an increased leakage current and a reduced parasitic capacitance. Therefore, process-independent retention time extension is required without the use of an additional capacitor in the eDRAM gain cell.

To overcome the retention time issue, 2T [21], 3T [22] and 4T [23] gain cell structures have been investigated. For comparison, they were implemented in the same 28 nm process, as shown in Table 1. The 2T and 3T gain cells have the advantage of a small area. However, their retention times are shorter than 100 ns, making them unsuitable for PIM applications. The retention time of the 4T gain cell was of the order of hundreds of microseconds [23] but was nonetheless insufficient to perform energy-efficient MAC operation in PIM applications without frequent refresh operations.

To solve these problems, this paper proposes a PS-GC eDRAM macro implemented in a 28 nm process. The PS-GC comprises a 2T gain cell and 3T pseudo-static leakage compensation (PSLC) that offers static charge storage by compensating the leakage current. The capacitors needed for retention time of data in a conventional eDRAM gain cell can be eliminated, improving the area efficiency and bit density of the eDRAM. The capacitor-free realization of the eDRAM allows analog PIMs to implement a charge-domain operation based on MOM capacitors. The proposed gain cell achieves area reductions of 21% and 42% compared to a 6T SRAM and 8T SRAM with the same process, respectively.

The rest of this paper is organized as follows. Section 2 describes the model and charge-loss mechanism of a conventional 2T1C gain cell. Section 3 presents a methodology to address retention time issue. Section 4 explains the proposed gain cell with PSLC. Section 5 presents the overall architecture of the eDRAM macro and simulation results. Finally, Section 6 presents the conclusions of this paper.

## 2. Conventional 2T1C Gain Cell and Charge Loss Mechanism

A conventional 2T1C gain cell [21] consists of two transistors and one capacitor, as shown in Figure 3. The transistors M1 and M2 control the write and read operations, respectively. For the data write operation, M1 is activated, and the voltage of the write bitline (WBL) is transferred to the internal storage node (SN). For the data read operation, M2 is activated, and the read bitline (RBL) is maintained at a precharged voltage or discharged depending on the stored data. 

During the data hold operation, the WBL and write wordline (WWL) are kept high (“1”). In case of the write operation, the WBL changes to “0” or “1” according to the data to be written, and the WWL is pulled down to “0”. During the read operation, the RBL is precharged to “1” before bit evaluation. The read wordline (RWL) is pulled down to “0” when WWL is kept high. If the value stored in the cell is data “1”, the RBL is discharged, and if it is data “0”, the RBL is maintained.

During the data hold operation, the retention time of data stored in SN is mainly determined by the capacitance C and leakage current via M1 and M2. The retention time can be extended by increasing the capacitance; however, a larger capacitor requires a larger area, and this reduces the bit density of the eDRAM. An overlaid MOM capacitor can be used to reduce the area overhead [11]; however, it cannot be applied to the eDRAM for PIM applications because many analog PIMs use the MOM capacitor for charge–domain computation. 

Therefore, the leakage current must be reduced to extend the data retention time. Charge leakage that results in data loss is caused by two reasons. First, assuming that the WBL and WWL are maintained high (“1”) during hold operation, the subthreshold current of M1 induces a current into the SN, and the voltage of SN can be increased by the leakage current of M1. If SN stores data “0”, the leakage current of M1 increases its voltage and the stored data flips to “1”, as shown in Figure 3. 

Second, the leakage current of M2 from RWL and RBL causes charge injection. If SN stores data “0”, the leakage current of M2 increases its voltage and the stored data flips to “1”. When storing data “1”, the charge stored at SN can be reduced owing to the leakage current during the read operation; however, the charge loss can be compensated by the leakage current of M1 during the standby period. As a result, stored data “1” has a longer retention time than that of stored data “0”, and the retention time of the gain cell is mainly determined by the case of storing ”0”.

Figure 4 shows the simulated leakage current waveforms after the write operation. In case of storing data “0”, a large amount of leakage current was injected after the write operation via M1, increasing voltage of SN. Compared with stored data “0”, the leakage current for stored data “1” was negligible, and the data was maintained without significant charge loss. 

Figure 5 shows the Monte Carlo simulation results of SN voltage during the data hold mode. The leakage current via M1 increases the SN voltage, resulting in the bit flipping from “0” to “1”. On the other hand, when the SN stores data “1”, the voltage was maintained without significant change because the leakage current via M1 maintains SN voltage. Thus, in order to improve the gain-cell usage in PIM applications, the retention time for storing data “0” must be extended by compensating the leakage current of M1.

## 3. Methodology

As discussed in the previous Section 2, bit cells in conventional eDRAMs may cause data flipping instantly after the write operation due to a sub-threshold leakage current, which can be more severe in deep submicron technology [24,25,26,27,28]. Various methods have been suggested to address those issues; however, those approaches resulted in significant area overhead or needed additional voltage boosting circuits. For example, additional bit line or word line were used to reduce the leakage current during data hold mode [29,30]. A boosted word line control voltage was used to reduce the sub-threshold current [23,31]. A large capacitor was used to be robust against the leakage current. The retention time could be extended by the previously suggested methods; however, a refresh operation was still required due to the finite retention time [32,33].

Instead of the previous approaches, this work focused on actively compensating the leakage current in the gain cell to prevent the stored data loss. The pseudo-static operation of the active leakage compensation offers much extended retention time compared with those of the previous eDRAM structures. The active leakage compensation can be less dependent to the process scaling using a feedback configuration. The proposed method can be implemented without uses of additional capacitors or boosted bit line or word line voltages. Hence, it is possible to realize both retention time extension and compact implementation by minimizing area overhead for the active leakage compensation.

To demonstrate the effectiveness of this work, the proposed gain cell and its 64 × 64 eDRAM macro were implemented with 28 nm CMOS process. They were verified based on post-layout Monte Carlo simulations considering process, voltage and temperature (PVT) variations. Simulation results of bit cell voltage after the write operation will be discussed to verify the pseudo-static operation. Simulated write-access time and read-access time of the eDRAM macro will be explained to demonstrate fast operation of the proposed gain cell. To verify robust operation of the eDRAM without data loss across the PVT variations and operating frequency conditions, Monte Carlo simulations with 1000 trials at each operating condition were conducted, being represented in Shmoo plots.

## 4. Operating Principle and Circuit Implementation of Proposed PS-GC

The proposed PS-GC can address the leakage issues in the conventional 2T1C gain cell by actively compensating the leakage current and can extend the retention time without requiring capacitors. Figure 6 shows its schematic. The PS-GC comprises a 2T gain cell and the PSLC. As with the above-described 2T1C gain cell, M1 and M2 are for write and read access control devices, respectively. The PSLC is activated when SN stores data “0”. By pulling down the leakage current through the PSLC, SN can hold data “0” without increasing the voltage. 

The PSLC is turned off when SN stores data “1”. As the PSLC performs active leakage compensation, the PS-GC can maintain the stored data without requiring the use of a large capacitance. Further, M1 can be implemented with low-*V**_TH_* (LVT) devices for fast write-access time without concerns over the leakage current. Therefore, the write-access time of the eDRAM can be improved without use of a voltage-boosted WWL control as in [23,31]. 

The PSLC is implemented with an auxiliary inverter with M3 and M4 and pull-down NMOS M5 for compensating the leakage current via M1. When SN stores data “0”, the auxiliary inverter turns on M5 to eliminate the leakage current injected into SN. As M5 forces SN to be zero, the PS-GC can maintain data “0” without the voltage increase. To minimize the subthreshold leakage current during the storage of data “1”, M5 is configured with the high-*V**_TH_* (HVT) device. 

The NMOSs in the PS-GC are implemented with the minimum width and length available in 28 nm CMOS process to minimize the bit cell area. In case of the PMOSs (M1, M3), larger width of 200 nm was used for accelerating write operation and charge leakage compensation because the PS-GC should quickly turn on M5 when the leakage current via M1 increases the SN node voltage when data “0” is stored in SN. 

Figure 7 shows the details of the PSLC operation in the PS-GC. If SN stores data “0”, M3 turns on and M4 turns off, thereby making M5 compensate the leakage current. Then, M5 forces the SN node to be zero by pulling down the residual charge injected through M1 and M2. This compensates the leakage current and prevents bit flipping. Accordingly, the PSLC effectively extends the retention time for storing data “0”, realizing pseudo-static operation of the gain cell. If SN stores data “1”, only M4 turns on, and the feedback path (FP) node is forced to be zero. 

Therefore, M5 turns off, and SN can maintain its voltage without significant charge loss. As M5 is implemented with a high-*V_TH_* device, the leakage current during deactivation is negligible, to compared with the charge injection via M1 and M2. In the case of a pull-down leakage current during the storage of data “1”, the charge loss in SN after the read or write operation is compensated automatically during the data hold operation because the WWL, WBL, RBL and RWL are maintained at “1” at this time, thus supplementing the charge loss via M1 and M2. As a result, the gain cell with PSLC exhibits unlimited retention time as with that of SRAM without requiring the use of a capacitor.

Figure 8 shows the simulated SN voltage based on Monte Carlo mismatch simulations to verify that the stored data in SN can be maintained well without the charge loss or bit flipping. After writing data “0” or “1”, the PSLC successfully enabled the gain cell to maintain the stored voltage. Hence, the SN voltage after 1ms from the write operation does not show any disturbance or variation. Figure 9 shows the post-layout simulated current consumption of the PS-GC during hold operation with data “0”. Owing to the positive feedback configuration of the PSLC, the leakage compensation maintains the SN node voltage regardless of the process and temperature variations. As the dominant leakage source is PMOS M1, the process corners of SF and FF show the largest static current consumption at temperature of 85 °C.

Figure 10a shows one of the analog PIM examples based on the PS-GC eDRAM. A column-wise MAC operation can be realized by the PIM using analog-to-digital converters (ADCs). The stored weights in the eDRAM are multiplied with RWL-encoded input data and accumulated in the RBL. For example, the RBL will be discharged when both weight and input data are “1”. Depending on the number of RBL discharging cells, the voltage at the RBL will be determined as shown in Figure 10b. Then, the column-wise ADC converts the RBL voltage, corresponding to the MAC result.

## 5. Simulation Results and Discussion

To demonstrate the pseudo-static operation of the proposed gain cell, a 4 kb eDRAM macro was implemented as shown in Figure 11. This macro comprises a 64 × 64 gain cell array, WWL decoders, RWL decoders, sense amplifiers, WBL drivers and pre-charge drivers. Figure 12 shows layout area of the proposed 4 kb eDRAM macro and comparison of bit cell layouts of 2T GC, 3T GC, PS-GC, single-ported 6T SRAM and dual-ported 8T SRAM, all of which were implemented in a 28 nm CMOS technology. The proposed gain cell has a size of 0.434 µm × 0.66 µm (0.286 µm²). This area was 0.79- and 0.58-times that of the 6T SRAM [34] and 8T SRAM [35], respectively. Therefore, eDRAM with the proposed gain cell can exhibit improved area efficiency and bit density, making it more suitable for use in IoT sensor nodes.

Figure 13 shows the post-layout simulated write-access time at five process corners and four temperature conditions. For writing data “1”, the eDRAM achieved write-access time below 0.5 ns for all process corners and temperature conditions. For writing data “0”, the eDRAM showed worst write-access time of 0.92 ns at the FS process corner and temperature of −25 °C. As the write operation is performed by the PMOS transistor (M1), writing time is faster when storing data “1” than that of storing data “0”. 

Figure 14 shows the post-layout simulated write access delay for supply voltages of 0.9–1.2 V. For the typical case (TT 25 °C), best case (SF 85 °C) and worst case (FS −25 °C), the eDRAM achieved a write-access time below 1 ns across the entire supply voltage range. As the PSLC allows the write access transistor (M1) to be implemented with a low-*V_TH_* PMOS without concern of increased leakage current, the PS-GC can achieve a fast write-access time.

Figure 15a shows the post-layout simulated read-access times with the five process corners and four temperature conditions. The read-access time included the evaluation delay at the sense amplifier. The eDRAM achieved read-access time below 0.5 ns for all process corners and temperature conditions. As the read operation in the gain cell is performed using NMOS transistor (M2), the worst read-access time was observed at the SS and SF process corners. Figure 15b shows the post-layout simulated read-access time with voltage range from 0.9 to 1.2 V. At a supply voltage of 1.2 V, the eDRAM can provide fast read-access time below 0.2 ns for the typical, best and worst process corners.

To demonstrate the eDRAM operation under various operating conditions, post-layout Monte Carlo mismatch simulations with 1000 trials were conducted, as shown in Figure 16. With operating frequencies of 100–667 MHz and TT, SF and FS process corners, the eDRAM operation was evaluated under supply voltages of 0.5–1.2 V and temperatures of −25 to 85 °C. At this time, the SF and FS process corners were used in consideration of the worst performances for the read and write access operations. Under each condition, the eDRAM operation was marked as “Fail” if one or more Monte Carlo trials, including bit flipping, failed to operate normally. 

At an operating frequency of 100 MHz, the eDRAM can operate with supply voltages higher than 0.7 V regardless of the process corners and temperatures. At an operating frequency of 250 MHz, the eDRAM maintains normal operation down to a supply voltage of 0.8 V. The maximal operating frequency of the eDRAM was 667 MHz. At this frequency, the eDRAM achieved an operating voltage range of 0.9–1.2 V across the entire temperature range and the three process corners. Shmoo plots demonstrated that the proposed gain cell and its 4 kb macro provide a wide operating range and high reliability while overcoming the retention time issues faced in the conventional eDRAM macro.

Table 2 summarizes the performance of the proposed gain cell and provides a comparison with previous gain cells [20,21,22,23,29,30]. The proposed gain cell had a compact area owing to the use of process scaling; further, it had unlimited retention time owing to the use of the PSLC. It does not require an additional bitline or wordline and boosted control voltage. The proposed gain cell and its eDRAM macro can be used effectively in PIM applications. Table 3 shows a fair performance comparison with 6T SRAM [34], 8T SRAM [35] and other eDRAM gain cells [21,22] implemented in a 28 nm process under the same process conditions. 

Compared with the 6T SRAM and 8T SRAM, the proposed gain cell offers a compact implementation with a more than 20% area reduction. Further, the static power of each bitcell in the proposed eDRAM is lower than those in 6T SRAM and 8T SRAM. Compared with the 2T and 3T gain cells, the proposed PS-GC dissipates a little bit more static current due to the pseudo-static operation. 

However, considering the unlimited retention time advantage of the PS-GC, the static current increase can be tolerable to many applications including the PIMs. As a result, the proposed gain cell and its eDRAM macro can offer both a compact realization and pseudo-static bit storage without a significant increase in the power consumption. Therefore, PIM using the proposed gain cell can be leveraged to realize improved area and power efficiency compared with SRAM-based PIM.

## 6. Conclusions

This paper presented a PS-GC for eDRAM to overcome the retention time issue faced in conventional eDRAMs. The PS-GC employs a PSLC in addition to a 2T gain cell without the use of a capacitor. The PSLC compensates for a leakage current injected into the PS-GC, preventing charge loss or bit flipping in the gain cell. The active leakage compensation of the PSLC successfully extends the retention time even in deep-submicron processes showing large leakage currents. 

The PS-GC achieves fast write-access time and read-access time without the concern of an increased leakage current. Thus, the PS-GC realizes not only compact and fast readout operations but also unlimited retention time as in SRAM operations. The elimination of the capacitor in the PS-GC improved the utility for analog PIMs by making room for overlaying capacitors on the memory cell for charge-domain computations. The PS-GC and its 64 × 64 eDRAM macro were implemented in a 28 nm process. 

The post-layout simulation results demonstrate that the eDRAM can operate under severe operating conditions with varying process corners, voltages of 0.9–1.2 V, temperatures of −25 to 85 °C and operating frequencies up to 667 MHz. Compared with previous eDRAM gain cells, the proposed gain cell exhibited a greatly extended retention time without increase in area, additional bitline or wordline, boosted control voltage. The PS-GC has lower static power consumption and a compact implementation compared with those of SRAMs. 

The implication of the PS-GC and its eDRAM macro can be discussed in terms of memory macro and analog PIM applications. The eDRAM with the PS-GC offers compact implementation of pseudo-static memory that can be applied to various system-on-chips (SoCs). If the SoCs require high bit density of embedded memory without refresh operation, the PS-GC and its eDRAM macro will be a good candidate instead of the SRAM. In case of the analog PIM applications, the PS-GC can leverage the processing efficiency of the PIM because of its capacitor-less compact design and decoupled fast read/write operation. Therefore, the implementation of the analog PIM using the PS-GC eDRAM will be in promising future work.

## Figures and Tables

**Figure 1 sensors-22-04284-f001:**
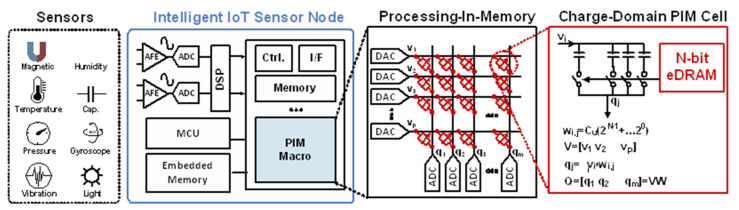
Conceptual block diagram of eDRAM-based PIM used for intelligent IoT sensor nodes.

**Figure 2 sensors-22-04284-f002:**
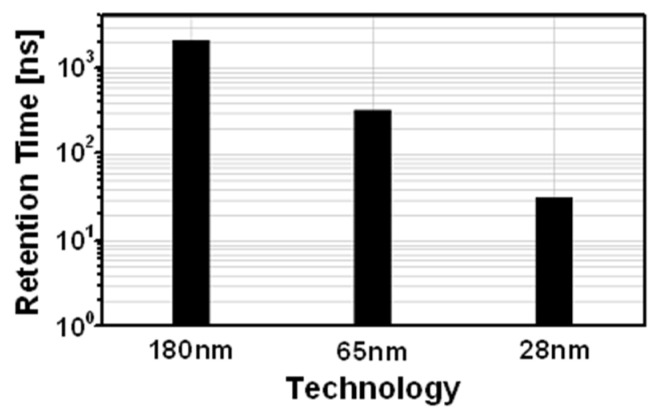
Retention time of 2T gain cell implemented in 28, 65 and 180 nm processes.

**Figure 3 sensors-22-04284-f003:**
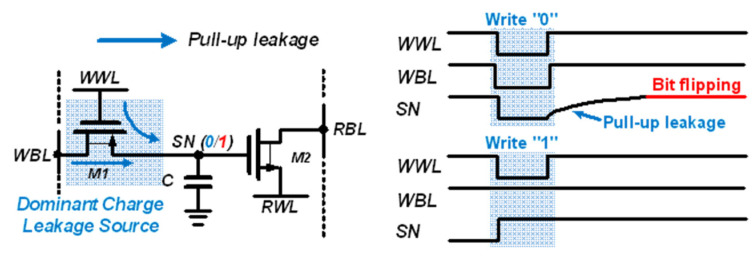
Schematic of conventional 2T1C gain cell.

**Figure 4 sensors-22-04284-f004:**
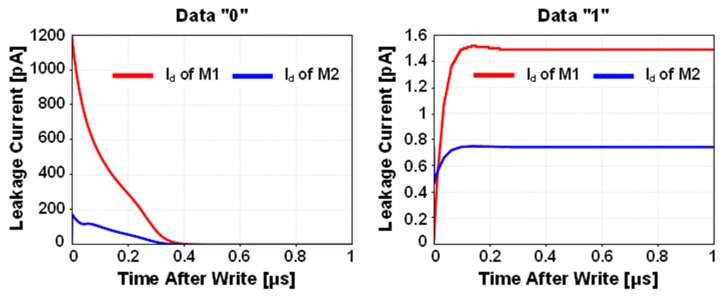
Simulated leakage current after write operation of 2T1C gain cell.

**Figure 5 sensors-22-04284-f005:**
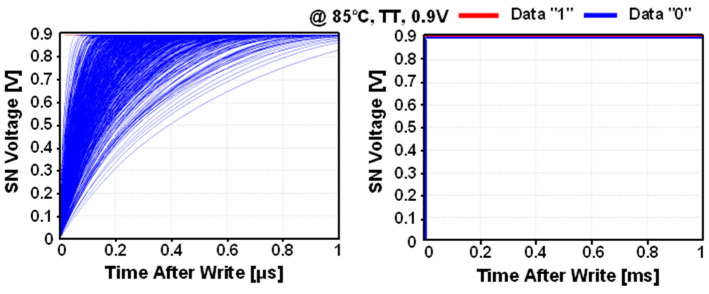
Monte Carlo simulation results of storage node (SN) voltage during the data hold mode of the 2T1C gain cell.

**Figure 6 sensors-22-04284-f006:**
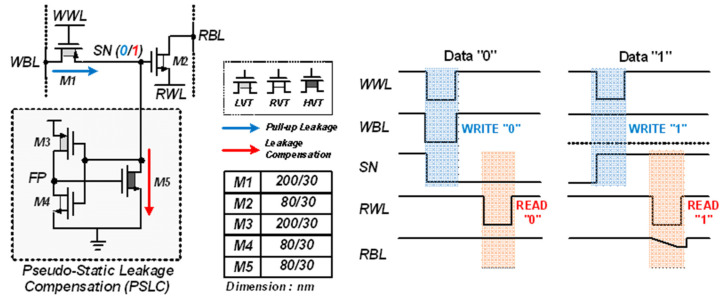
Schematic and conceptual timing diagram of proposed PS-GC.

**Figure 7 sensors-22-04284-f007:**
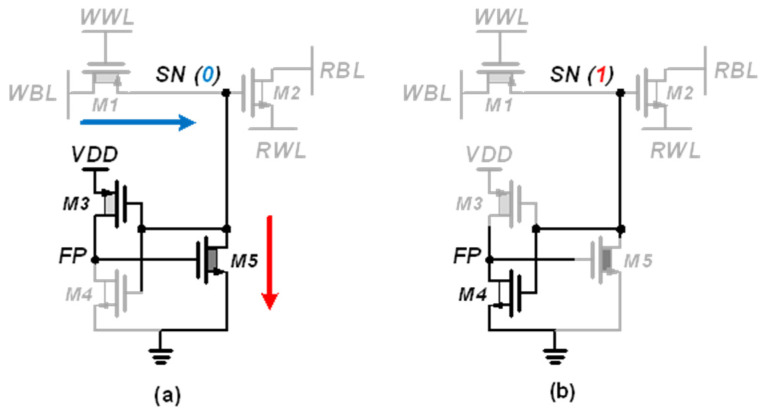
Leakage compensation of PS-GC when storing data (**a**) “0” and (**b**) “1”.

**Figure 8 sensors-22-04284-f008:**
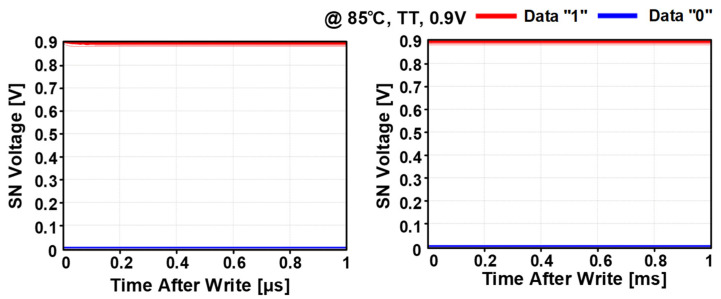
Monte Carlo mismatch simulation of data retention after storing data “0” and “1”.

**Figure 9 sensors-22-04284-f009:**
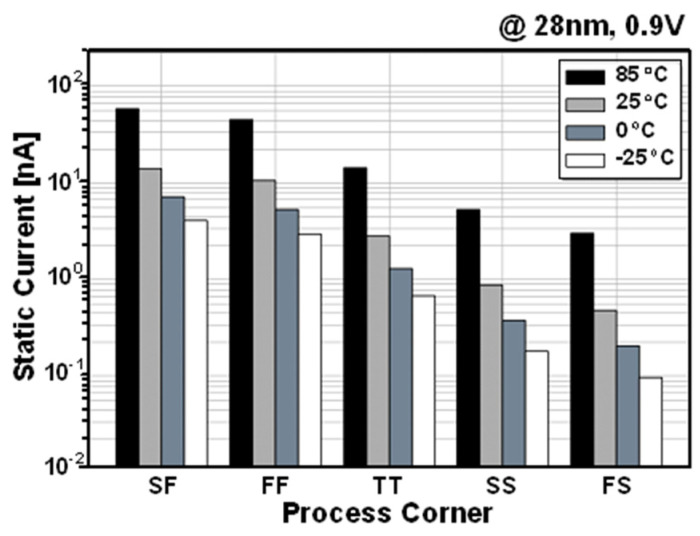
Post-layout simulated static current of PS-GC.

**Figure 10 sensors-22-04284-f010:**
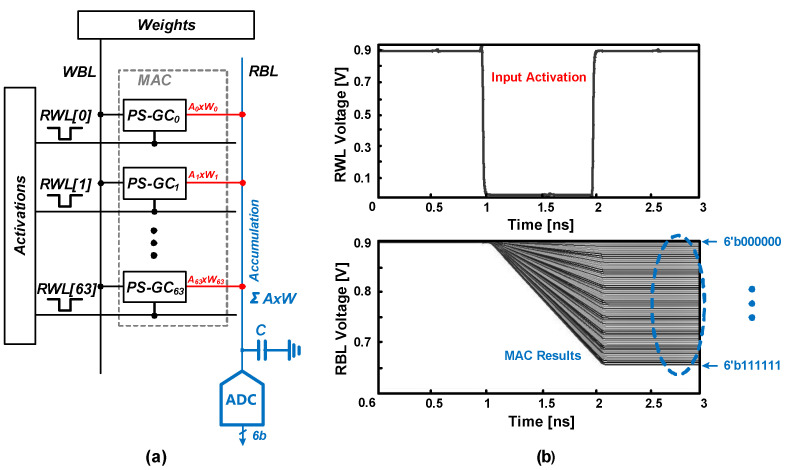
(**a**) PIM configuration example of PS-GC eDRAM. (**b**) RBL discharge plot of all accumulation results.

**Figure 11 sensors-22-04284-f011:**
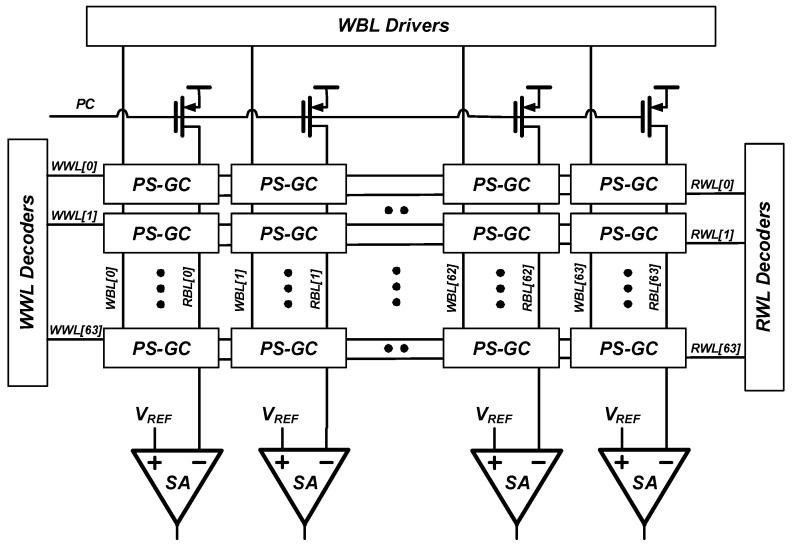
Overall architecture of the 4 kb eDRAM macro.

**Figure 12 sensors-22-04284-f012:**
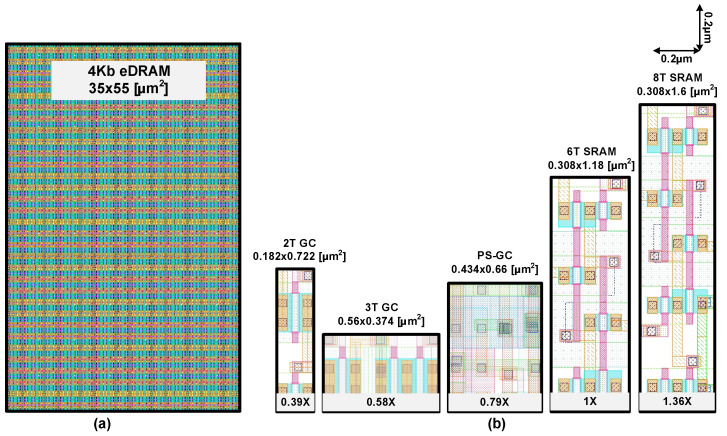
(**a**) Proposed 4 kb macro layout. (**b**) Layout comparison of bit cells of eDRAMs and SRAMs.

**Figure 13 sensors-22-04284-f013:**
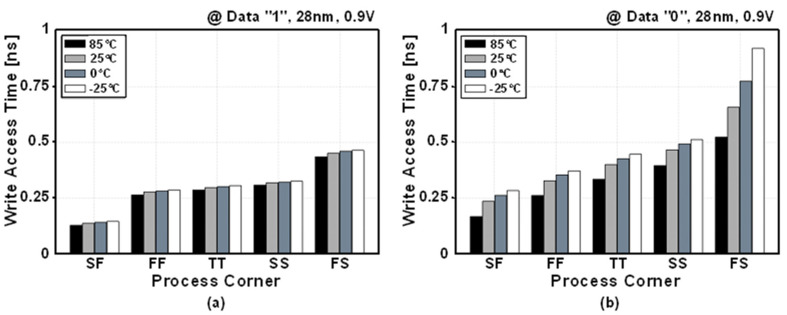
Post-layout simulated write-access times when storing data (**a**) “0” and (**b**) “1” across five process corners and four temperature cases.

**Figure 14 sensors-22-04284-f014:**
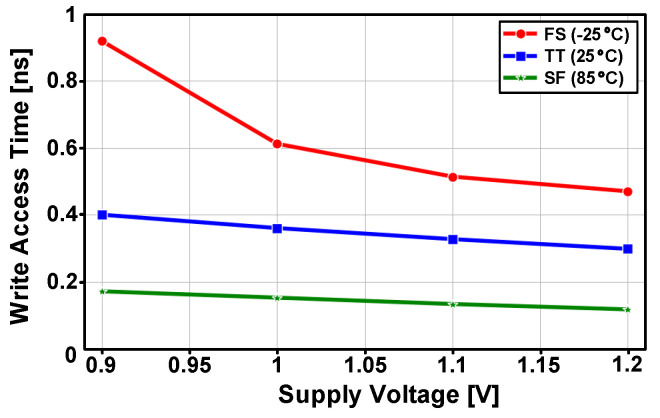
Post-layout simulated access time versus supply voltage with typical (TT 25 °C), best (SF 85 °C) and worst (FS −25 °C) process corners and temperature conditions.

**Figure 15 sensors-22-04284-f015:**
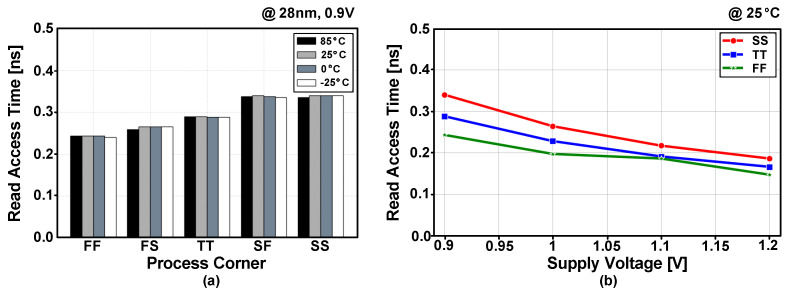
Post-layout simulated read-access times (**a**) depending on process corners and temperatures and (**b**) across the supply voltage range with typical (TT), best (FF) and worst (SS) process corners.

**Figure 16 sensors-22-04284-f016:**
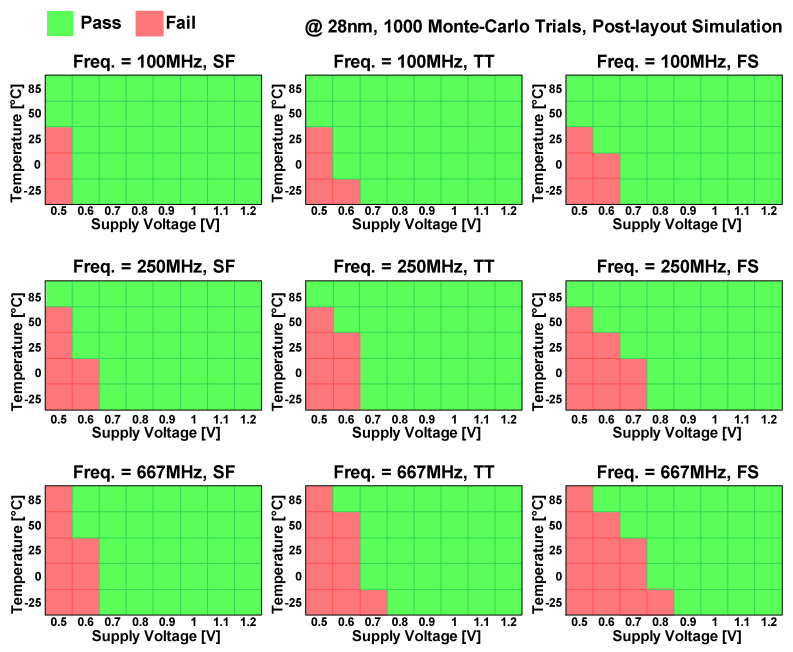
Shmoo plots of post-layout Monte Carlo 1000-trial simulations of the eDRAM with various operating frequencies (100–667 MHz), process corners (SF, TT and FS), temperatures (−25 °C to 85 °C) and supply voltages (0.5–1.2 V).

**Table 1 sensors-22-04284-t001:** Comparison of the retention times of eDRAM gain cells implemented in a 28 nm process.

Structure	2T [21]	3T [22]	4T [23]
Cell size (µm^2^)	0.14	0.209	0.25
Retention time	22.4 ns	50 ns	154 µs
Storage	MOS gate (<1 fF)	MOS gate (<1 fF)	MOS gate (<1 fF)
Supply voltage	1.1 V	1.2 V	0.7 V
Process	28 nm (Converted)	28 nm (Converted)	28 nm FD-SOI

**Table 2 sensors-22-04284-t002:** Performance summary and comparison with previous works.

	2T [20]	2T [21]	3T [22]	3T [29]	3T [30]	4T [23]	This Work
BitcellSchematic	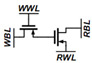	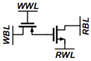	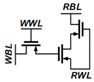	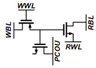	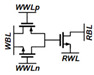	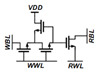	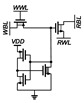
Process	65 nm	65 nm LP	65 nm LP	65 nm LP	180 nm	28 nm FD-SOI	28 nm
Bitcell Size	0.275 μm^2^	0.478 μm^2^	0.627 μm^2^	0.674 μm^2^	6.97 μm^2^	0.23 μm^2^	0.286 μm^2^
Retention Time	10 μs@ 85°C	276.5 μs@ 85°C	1.25 ms@ 85°C	325 μs@ 85°C	0.8 ms@ 27°C	154 μs@ 85°C	Static
Maximum Freq.	2 GHz	667 MHz	NA	1 GHz	40 MHz	66 MHz	100 MHz	667 MHz
V_DD_ Range	0.7–1.1 V	0.8–1.4 V	0.8–1.3 V	0.8–1.2 V	0.6–1.2 V	0.6–0.9 V	0.7–1.2 V	0.9–1.2 V
Temp. Range	25–85 °C	25–85 °C	25–85 °C	25–85 °C	27 °C	0–85 °C	−25–85 °C
Write AccessTime	NA	0.21 ns@ 85 °C	0.27 ns@ 85 °C	1.5 ns@ 85 °C	1.3 ns@ 27 °C	0.46–0.67 ns@ 27 °C	<0.3 ns@ 85 °C, TT
Read AccessTime	NA	0.46 ns@ 85 °C	0.61 ns@ 85 °C	1 ns@ 85 °C	NA	<3 ns@ 27 °C	<0.3 ns@ 85 °C, TT
AdditionalBit/Word Line?	No	No	No	Yes	Yes	No	No
Need Boosted Voltage?	Yes	Yes	Yes	Yes	No	Yes	No
Need Refresh?	Yes	Yes	Yes	Yes	Yes	Yes	No
Retention Power	508 mW/2 Mb@ 85 °C	1.16 mW/Mb@ 85 °C	1.25 mW/Mb@ 85 °C	NA	1 μW/2 kb@ 27 °C	909 nW/8 kb@ 85 °C	22.5 μW/4 kb@ 85 °C, TT

**Table 3 sensors-22-04284-t003:** Comparison of memory cells implemented in the 28 nm process.

	6T SRAM [34]	8T SRAM [35]	2T eDRAM [21]	3T eDRAM [22]	This Work
BitcellSchematic	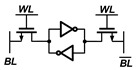	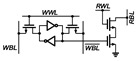	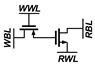	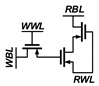	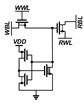
Process	28 nm	28 nm	28 nm	28 nm	28 nm
Normalized Area	1 X	1.36 X	0.39 X	0.58 X	0.79 X
Bitcell Size	0.36 μm^2^	0.491 μm^2^	0.14 μm^2^	0.209 μm^2^	0.286 μm^2^
Retention Time	Static	Static	17.2 ns	42.1 ns	Static
Bitcell Access	Shared R/W	Decoupled R/W	Decoupled R/W	Decoupled R/W	Decoupled R/W
Storage	Latch	Latch	MOS gate (<1 fF)	MOS gate (<1 fF)	MOS gate (<1 fF)
Supply Voltage	0.9 V	0.9 V	0.9 V	0.9 V	0.9 V
Static Power/bit	50 nW	69 nW	0.16 nW	1.51 nW	FS (−25 °C)	TT (25 °C)	SF (85 °C)
0.07 nW	2.2 nW	47.4 nW

## Data Availability

Not applicable.

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
