# Peer review of "Pseudo-Static Gain Cell of Embedded DRAM for Processing-in-Memory in Intelligent IoT Sensor Nodes"

_sensors, 2022, doi:10.3390/s22114284_

Round 1
Reviewer 1 Report
- The idea of the research topic is excellent but poor representation
- So many references are mentioned in the "Introduction" section while you can make a separate section named "Related Work/Literature Review" and put as many recent works as you can.
- No clear Methodology was found and mentioned at the time of reviewing your article. You should add a separate section named "Methodology" and should mention your entire method and purpose here as the purpose of your paper is not also clear by your description.
- Tables 2 and 3 are not actually tables, those are screenshots/pictures. You should take care of it next time. If you mention a Table then you should provide a proper Table not a picture or anything else. Also, the quality of the figure is very low, please provide high-quality figures
- The conclusion is not clear and please add some future works in the conclusion or make a separate section named "Future Work" and add some future commitment to implement.
Author Response
The authors really appreciate the valuable comments from the reviewers for the revised manuscript. We carefully reviewed and revised the manuscript to address all the concerns, and we believe the paper became much better thanks to the reviewers’ contribution. We greatly appreciate your thoughtful attention and support for our manuscript. The attached materials are the responses to the reviewers’ comments. In the manuscript, we marked the revised text and figure as red color.

Reviewer 2 Report
- The contributions discussion in the introduction section could benefit from being more concise and clear, and this would quickly give the reader a better understanding of the key goals and advantages of the work.
- The list of references should be reformatted and checked again to be matched with the journal requirement where a different styles and types are used. Please check the some spells and typos
- The abstract must summarize the performance evaluation results.
- The results should be further analyzed, more details and further discussion of the simulation results is needed.
Author Response

(The authors gave the same response as above.)

Reviewer 3 Report
The authors have presented a pseudo-static gain cell for EDRAM macro which solves retention time issues in traditional designs. Extensive post-layout simulation results in 28nm process have been presented. There are significant improvements in retention capability of the memory cell compared to previous work. The paper is very well written and overall an interesting read. Few comments and suggestions:
in page 4, " Monte Carlos" should be "Monte Carlo".
in figure 6, how were the various transistor sizes (M1-M5) chosen? please provide more details of the rationale behind this choice.
in figure 11, are the layouts b,c,d in the same scale? might be a good idea to label the area differences (0.79x, 1x, 1.36x from table 3) in this figure.
please also include the layout of traditional 2T and 3T EDRAM cell in figure 11 for fair comparison.
how does power consumption of the proposed EDRAM cell compare with traditional 2T and 3T EDRAM cell designs? please provide simulation results, analysis / discussion and also include in table 3.
processing in memory has been mentioned as the target application in several places in the paper and also in the title. please include some more discussion on how the proposed pseudo-static EDRAM array can be used for such processing, at least with a small computation example and corresponding simulation results.
resolution of the figures must be improved. The figures are well drawn but many of them are blurred due to low resolution. Please revise accordingly.
Author Response

(The authors gave the same response as above.)

Round 2
Reviewer 1 Report
- Everything looks good and revised well as instructed in the previous review. - The concepts explained in the paper are crisp and clear.
- Overall the paper is good.
Reviewer 2 Report
Authors mention all comments and paper can be accepted in current version
Reviewer 3 Report
The manuscript is well revised and ready for acceptance.